# Opioid Reduced Anesthesia in Major Oncologic Cervicofacial Surgery: A Retrospective Study

**DOI:** 10.3390/jcm12030904

**Published:** 2023-01-23

**Authors:** Emma Evrard, Cyrus Motamed, Arnaud Pagès, Lauriane Bordenave

**Affiliations:** 1Department of Anesthesiology, Gustave Roussy, 94805 Villejuif, France; 2Faculty of Medicine, University of Paris-Saclay, 94270 Le Kremlin Bicêtre, France; 3Department of Biostatistics and Epidemiology, Gustave Roussy, 94805 Villejuif, France

**Keywords:** opioid free anesthesia, opioid reduced anesthesia, dexmedetomidine, cervicofacial oncologic surgery

## Abstract

Opioid sparing is one of the new challenges in anesthesia and perioperative medicine. Opioid reduced anesthesia (ORA) is part of this approach, and it consists of a multimodal analgesia-associating non-opioid analgesic regional anesthesia to reduce intraoperative opioid requirements. Major cervicofacial oncologic surgery could specifically benefit from ORA, since it is known to generate intense and prolonged postoperative pain, with a high risk of pulmonary complications. Methods: This is a retrospective case-controlled study of 172 patients with major cervicofacial oncologic surgery. Group ORA (dexmedetomidine and lidocaine), *n* = 86, was compared to patients treated with standard opioid based anesthesia, Group control, *n* = 86. The main endpoint was to study perioperative opioid consumption and postoperative pain scores, and the secondary endpoint was to observe opioid related side effects. Results: The ORA group received 6.2 ± 3.1 mg morphine titration at the end of surgery, while the control group received 10.1 ± 3.7 mg *p* < 0.0001; there was no significant difference in post-operative analgesia requirements and pain scores between the groups. Intraoperatively, the ORA protocol yielded bradycardia in 4 persons, while in the control group, only 2 persons had bradycardia necessitating intervention, *p* < 0.05. Postoperatively, episodes of hypoxemia (50%) and the need for additional pressure-assisted ventilation (6%), was significantly different in the ORA group than in the control group (70% and 19%), *p* < 0.05. There was no difference between the two groups for the incidence of nausea and vomiting, ileus, or postoperative delirium. Discussion: ORA was not associated with a decrease in postoperative pain and opioid requirement, but possibly reduced the incidence of hypoxemia and the use of additional pressure-assisted ventilation, although we cannot rule out confounding factors. The possible benefits of ORA remain to be demonstrated by prospective studies.

## 1. Introduction

Opioids are historically part of the fundamental tripod of anesthesia, in association with hypnotics and neuromuscular blockers. In France, 1.1% of the population received a prescription for strong opioids in 2017, with an increase of +104% between 2004 and 2017. According to pharmacovigilance reports, this increase in consumption was also accompanied by episodes of overdoses, which increased by 98% between 2004 and 2016 [1].

Opioids are frequently administered during the perioperative period; it is estimated that 50% of patients are discharged with a prescription for strong opioids for the management of postoperative pain, and more than 3% still use them 3 months later [2].

Opioids have multiple side effects, including a dose-dependent respiratory depressant effect, sedation, chest rigidity, cough depression, and bronchoconstriction at high doses. A total of 46% percent of patients treated with intravenous opioids experience respiratory depression [3]. Postoperative nausea and vomiting (PONV) is the most frequent and feared side effect of opioids, with a mean incidence of 25% for vomiting and 52% for nausea [4]. Finally, opioids may induce hyperalgesia, tolerance, and dependence. Hyperalgesia refers to an increased sensitivity to feeling pain from a stimulus that usually provokes it. In a meta-analysis by Fletcher et al., high-dose vs. low-dose intraoperative remifentanil was responsible for a significant increase in early postoperative pain scores and was associated with increased morphine consumption [5]. Tolerance refers to the decrease in a pharmacologic effect and the need’ to increase the dose required to achieve the same effect; it can occur during both occasional and chronic pain, which can subsequently complicate pain management and increase the risk of opioid-related adverse events [6].

New anesthetic strategies seek to rationalize the administration of opioids by considering new drug synergies. One alternative is opioid free anesthesia (OFA) or, more moderately, opioid reduced anesthesia (ORA). Under general anesthesia, the patient will not experience pain, but rather nociception, which is the propagation of a painful stimulus by the sensory system and the reflex activation of the sympathetic system. Therefore, the management of intraoperative analgesia corresponds to the control of the hemodynamic response to nociception [7]. The activation of opioid receptors is one pathway of blocking the transmission of nociceptive information, but it is not the only one. OFA considers the plurality of mechanisms of action involved in nociception and is based on a balanced and multimodal analgesia by combining regional anesthesia (RA), NMDA receptor antagonists (ketamine, magnesium sulfate), anti-inflammatory drugs (NSAIDs, dexamethasone, and intravenous lidocaine), and α-2 agonists (dexmedetomidine or clonidine). The concept of OFA/ORA uses this synergy of action on different receptors to counter the nociceptive response to minimize the use of opioids. To date, OFA is a controversial strategy, despite a recent meta-analysis describing postoperative outcome improvements in several surgical settings [8]. Nevertheless, its clinical value is still being evaluated, as there are only few robust studies in this field. In the worrying context of the opioid crisis, it remains a hot topic: 74 ongoing studies on OFA are listed on clinicaltrials.gov. Another recent meta-analysis of 23 randomized controlled trials and more than 1000 patients observed equivalent analgesia between patients who received opioids and those in the OFA group at 2 h postoperatively, with a 20% reduction in PONV in favor of OFA [9]. However, it did not observe a significant difference in postoperative morphine consumption [9]. Another recent meta-analysis by Salomé did not find any clinically relevant benefit to OFA in terms of analgesia or postoperative opioid consumption [10]. Less radical, ORA aims only to reduce the use of intraoperative opioids, without eradicating them completely.

Pain is the most frequent symptom related to cervicofacial oncologic surgery: 86% of patients describe pain at the time of diagnosis [11]. This pain is exacerbated postoperatively and uncontrolled in 50% of patients [12]. Cervicofacial cancer surgery is often a major, long, and decaying surgery. In the case of extended tumor resection, the need for a reconstruction flap to fill in the loss of substance makes it a double or even triple site surgery. While the flap harvest site is often accessible to regional anesthesia to limit postoperative pain, facial blocks to cover the tumor resection area are rarely performed in routine practice. The accumulation of these multiple sites is responsible for complex pain mechanisms in the postoperative period, which are difficult to relieve despite a quality multimodal systemic analgesia [13]. Moreover, cervicofacial cancer patients often present several risk factors for postoperative complications: alcohol and tobacco addictions, respiratory and cardiovascular comorbidities, malnutrition, and chronic pain [14]. A total of 41% percent of cervicofacial cancer surgery patients use opioids preoperatively [15]. All these vulnerabilities lead to a very high postoperative morbidity and mortality: 43% of patients present a respiratory complication after this type of surgery with free flap reconstruction, and 10% acquire a pulmonary infection after laryngectomy. Finally, postoperative hospital mortality is 4% in these patients vs. 1% in the general population [16]. To date, there is no study evaluating the use of OFA or ORA in cervicofacial oncologic surgery. In this retrospective study, we investigated whether intraoperative morphine sparing with ORA was associated with a better intra- and postoperative analgesia and a reduction in opioid-related side effects compared to traditional opioid-based anesthesia.

Our main endpoint was the intra- and postoperative opioid requirements and postoperative pain scores in major cervicofacial cancer surgery by using an ORA protocol.

## 2. Materials and Methods

This single center retrospective study was performed between January 2019 and March 2020. Patients data were collected and processed in agreement with Gustave Roussy institutional review board approval on 11 September 2020, which did not identify any element contrary to medical ethics. In accordance with the recommendations of the Commission Nationale de l’Informatique et des Libertés and the new European GRPD regulations, patients were informed of the collection of their data by an information letter and could object if they wished. The control group consisted of scheduled major cervicofacial surgery lasting more than 4 h, with or without reconstruction. The indications of these major surgeries were decided by a multidisciplinary committee; patients with cardiovascular conditions, respiratory instability, or cognitive disability, along with other vital emergency situations, were excluded until improvement and stabilization were achieved.

All patients of the ORA group exhibited the same indications and counter indications as the control group; however, patients were included if they had no counter indications to ORA medications, which were dexmedetomidine and IV lidocaine. These exclusion criteria were: patients with cardiac conduction disorders, such as atrioventricular block or sinoatrial block, patients treated with beta-blockers and calcium channel blockers, those with a heart rate lower than 50/min during the anesthesia consultation, and patients with severe malnutrition.

### 2.1. Anesthesia Protocol and Postoperative Management

#### Protocol

In all patients, general anesthesia included propofol titration, ketamine, dexamethasone, and a non-depolarizing neuromuscular blockade. Anesthesia was maintained with sevoflurane, desflurane, or total intravenous target-controlled anesthesia (TIVA) with propofol, depending on the patient’s medical history, and IV bolus (0.3 mg/kg) followed by 0.15 mg/kg/h of ketamine. A peripheral block with a bolus of ropivacaine 2% was performed before induction at the harvest site in case of reconstruction, when possible, to improve intra- and postoperative analgesia [17].

In the control group, remifentanil was administered by target-controlled infusion (TCI), whereas in the ORA group, intraoperative analgesia was provided by a mean IV bolus of dexmedetomidine 0.4µg/kg at induction, followed by a continuous infusion at the discretion of the anesthetist in charge. Lidocaine IV was started with a bolus of 1.5 mg/kg for patients not treated with regional anesthesia, followed by a continuous infusion of 1 mg/kg/h for all patients, which was stopped at the start of skin closure. In the ORA Group, TCI remifentanil was still connected as a back-up, but administered only if the hemodynamic response to nociception, defined as tachycardia or hypertension, did not appear to be controlled by the ORA protocol alone. In both groups, intraoperative changes in remifentanil targets and dexmedetomidine doses were at the discretion of the anesthesiologist in charge. All patients were administered an infusion of magnesium sulfate (2 g) intraoperatively.

In both groups, multimodal postoperative analgesia at the end of the procedure included paracetamol, nonsteroidal anti-inflammatory drugs, nefopam, and a morphine titration (0.5–0.15 mg/kg) before awakening, followed by intravenous morphine in patient controlled analgesia mode (PCA) for 24 to 72 h. A continuous perineural infusion of ropivacaine was prescribed in patients who benefited from a regional block with peri-neural catheterization. Intraoperative monitoring of the patients included invasive measurement of blood pressure by arterial catheter associated with a pulse wave contour analysis system (EV1000^®^) Edwards Lifesciences Corp., Irvine, CA, USA, monitoring of the depth of anesthesia by the bispectral index, and monitoring of the neuromuscular blockade using an NMT Philips Intellivue accelerometer module (Royal Philips Electronics, Amsterdam, The Netherlands).

Postoperatively, patients were transferred to the post-anesthesia care unit (PACU) and then to the surgical continuous care unit (SCCU) for 24 to 72 h, depending on the surgery and the evolution. Patients with hypoxemia or desaturation less than 95% necessitating more than 3L of oxygen, or other type of mild respiratory complications, such as atelectasis, could benefit from high-flow nasal oxygen therapy or intermittent pressure-assisted ventilation support. The intensivist in charge could also decide the need for continuous invasive ventilation at any time, and transfer the patient to a medical intensive care unit for more respiratory support if needed.

In addition, from November 2019, patients with tracheostomy at the end of surgery were also included in a preemptive respiratory optimization protocol, with pressure-assisted ventilation (PAV) as part of a quality assurance program. This protocol consisted of 1 session of 30 min of PAV 6 times a day for 24 h, starting in the PACU, and continued in the SICU. In this protocol, FiO_2_ was adapted to have a saturation above 95%, and pressure ventilation was adjusted to obtain a tidal volume of 6–8 mL/min.

The following main endpoint parameters were recorded; intraoperative remifentanil dose and morphine titration dose at the end of the operation, as well as at Day 1 and Day 3, and the occurrence of uncontrolled pain (defined by a numeric pain rating verbal scale (NVS) > 3) and clinically acceptable pain (defined by a NVS ≤ 3) during the first 72 postoperative hours, as well as the site of origin of the pain.

The secondary parameters were: episodes of postoperative hypoxemia defined as SpO_2_ < 95%, or the need of oxygen higher than 3 L/min. The necessity of additional pressure-assisted ventilation or high-flow nasal oxygen postoperatively, in the case of hypoxemia or hypoventilation, the occurrence of postoperative nausea and vomiting (PONV) until Day3, postoperative ileus defined by the absence of stool at D3, urinary retention defined by the necessity of a new bladder catheterization after removal of the urinary catheter, and post-operative delirium defined by an equivalent score on the Nursing Delirium Screening Scale (Nu-DESC) > 2.

Hemodynamic tolerance of the anesthetic protocol was assessed by episodes of bradycardia requiring a bolus of atropine and intraoperative hypotension evaluated by the average noradrenaline flow in mg/h.

### 2.2. Statistical Analysis

Patient data were anonymized and recorded in the REDCap^®^ database (Vanderbilt University, Nashville, TN, USA) of our institution. Qualitative variables were described using numbers and percentages. Quantitative variables were presented by their mean and standard deviation, when the distribution was normal, and by their median and interquartile range (25th and 75th percentile of the distribution) otherwise.

To test the association between anesthesia technique and the different qualitative variables of interest, we used the Chi-squared test, if validity conditions were met, and the Fisher exact test otherwise. For quantitative variables, we used the nonparametric Wilcoxon test.

Linear regression was employed to adjust the effect of ORA on opioid consumption (intraoperative remifentanil dose, morphine titration dose at the end of the operation, and morphine consumption at Day 1 and Day 3). The following variables were used in the adjusted models: sex, age (years), and ASA score groups (ASA score I and II, ASA score III and IV).

All statistical analyses were performed with SAS© 9.4 software (SAS Institute Inc., Cary, NC, USA).

## 3. Results

### 3.1. Patient Characteristics

Data from 172 patients were collected in the analysis, with 86 patients in each group. The main demographic characteristics were comparable: 39.6% were women, with a mean age of 58 years. A total of 19.2% of patients were in chronic pain and were receiving an opioid treatment preoperatively. However, the two groups were not comparable regarding all characteristics (Table 1).

Surgeries lasted a median of 10 h (minimum = 4; maximum = 16) and consisted mainly of mandibulectomy and pharyngectomy with flap reconstruction. Tracheotomized patients in the ORA group (71 out of 86) received postoperative preemptive pressure support ventilation as part of a quality assurance program. In the control group, 82 out of 86 benefited from this protocol.

Main endpoint: opioid consumption and pain scores.

The ORA group received significantly less remifentanil intraoperatively 0.01 µg/kg/min vs. 0.07 µg/kg/min in the historical cohort (*p* < 0.0001). The morphine titration dose at the end of the procedure was also significantly lower in the ORA group: 6.2 ± 3.1 vs. 10.1 ± 3.7 mg in the control group (*p* < 0.0001)**.**

Postoperatively, the cumulative consumption of morphine by PCA at D1 and D3 was similar between the 2 groups, respectively, with 18mg in median at D1(min = 4; max = 20) in the control group vs. 17 (min = 6; max = 30) mg in the ORA group (*p* = 0.639) and 34 (18–63) vs. 38 (16–73) mg at D3 (*p* = 0.799).

Pain scores: At Day1, the incidence of clinically acceptable pain relief at rest (NVS < 3) was significantly higher in the OFA group, with 48.8% vs. 29.4% in the control group, (*p* = 0.009). No significant difference was observed for uncontrolled pain.

No difference was noticed at Day 2 and Day3.

Intraoperative events and drugs used are displayed in Table 2.

The pain score assessment was equivalent between the two groups over the 72 h (Figure 1).

Pain was mainly localized at the cervical and facial area (81%), followed by the flap harvest site (43%), and then the tracheostomy (26%) (Figure 2).

### 3.2. Secondary Endpoints

#### Opioids Adverse Effects Endpoints

The ORA group experienced 43 (50%) episodes of hypoxemia, while the control group exhibited 60 (70%) (*p* = 0.013). The necessity of using additional PAV or HFO was lower in the ORA group, with 5 (6%) vs. 16 (19%) in the control group (*p* = 0.018).

The incidence of PONV at Day2 was similar in both groups, with 17 (20%) in the ORA group vs. 16 (19%) in the control group (*p* = 0.999).

Hemodynamic tolerance of the anesthetic protocol: the rate of infusion of vasopressor support by norepinephrine was significantly more important in the ORA group in comparison to the control group, with a mean of 0.2mg/hour vs. 0.1mg/hour (*p* = 0.044). Other secondary endpoints are displayed in Table 3.

## 4. Discussion

In this retrospective study, ORA protocol did not have a significant impact on pain scores or postoperative morphine consumption, despite a reduction in intraoperative opioid doses. Pain was not optimized in more than 50% of the patients, underlining the difficulty of postoperative analgesic management in major cervicofacial cancer surgery patients [17]. Indeed, this is a surgery involving a highly innervated anatomical region. The pain trajectory of cervicofacial cancer patients is complex and might be characterized by paroxysmal attacks of pain with a continuous pain background, associating neuropathic, bone, joint, and cutaneous-mucosal and multi-site pain with a significant inflammatory component [13,18]. The resection surgery most often requires a flap covering, adding another pain site. Although the flap harvest site pain can be mostly relieved by regional anesthesia, cervicofacial blocks, such as those involving the V2 and V3 (trigeminal) nerves, are practiced by only a few teams in routine clinical practice; these procedures should be developed and their effect on acute and possible chronic pain studied. ORA is a multimodal anti-nociceptive strategy with an anti-inflammatory component, achieved by intravenous lidocaine; however, it seems difficult to prejudge its effectiveness in such a painful surgery where patients are often pre-exposed to opioids.

Concerning the adverse effects of opioids, ORA patients showed a statistically significant reduction in hypoxemic events, as well as postoperative PAV or HFO; we do not believe this is attributable to the lesser opioid use in the ORA group at Day 1, since there is not enough data in this retrospective study to speculate further on this result, as our respiratory PAV protocol might also have been a confounding factor in this small-sized heterogeneous population.

There was no reduction in other opioids side effects, such as PONV, in the ORA group. In addition, we noticed a particularly smooth awakening in the ORA group that persisted for 24 h after surgery. However, this observation was subjective, since there was no standardized planned evaluation to compare the two groups in this respect. Alpha-2 agonists are recognized and used daily in intensive care for their sedative and analgesic virtues in assisted ventilation weaning and in the prevention of delirium [19].

The cardiac rhythm tolerance of the ORA protocol was acceptable, with the occurrence of 4 episodes of bradycardia requiring atropine. There was also some intraoperative hypotension requiring vasopressor support in the ORA group, probably related to the vasoplegia induced by dexmedetomidine and intravenous lidocaine.

Major cervicofacial oncologic surgery is characterized by a high intraoperative blood pressure lability. It includes an initial period of tumor debulking, marked by a major nociceptive stimulation associated with a hemodynamic response, which decreases as soon as the tumor is resected. Subsequently, the blood pressure maintenance objectives shift to focus on the perfusion of the free flap. This blood pressure lability is also related to the vasculopathy of cervicofacial cancer patients, some of whom have lost the carotid baroreflex following previous cervical radiotherapy and have sequential post-radiation dysautonomia.

To develop perioperative medicine, the introduction of the ORA protocol in our department was part of a global approach to improve recovery after major surgery. The concomitant implementation of a protocol of respiratory rehabilitation by preemptive PAV and the surge in practice of regional anesthesia at the harvest site may have been confounding factors.

Data in the literature on OFA and ORA are discordant. Mulier’s randomized controlled trial described a decrease in postoperative pain, opioid consumption, desaturations, and PONV in the OFA group vs. anesthesia with opioids in laparoscopic bariatric surgery, with no difference in intraoperative hemodynamics [20]. Similarly, the randomized controlled trial of 80 bariatric laparoscopic urological surgery patients by Bhardwaj et al. revealed fewer respiratory depressions and better analgesia in the OFA group [21]. No episodes of bradycardia were described in this study.

On the other hand, the recent randomized controlled trial POFA of 303 patients, led by Beloeil et al., was discontinued prematurely because of episodes of severe bradycardia attributed to dexmedetomidine [22]. Unexpectedly, more respiratory events were found in the OFA group. There was no difference in postoperative pain, but there was a decrease in opioid consumption. The OFA group exhibited less PONV, but there was no difference in postoperative ileus. The primary endpoint in this latter study was a composite including hypoxemia, nausea-vomiting, and postoperative cognitive dysfunction as adverse effects of opioids, possibly losing specificity upon statistical evaluation.

The hemodynamic adverse effects of alpha 2 agonists in anesthesia were confirmed by the meta-analysis conducted by Demiri et al., which included more than 56 studies and 4800 patients. Indeed, they were significantly associated with more hypotensive episodes and bradycardia, both pre- and postoperatively [23]. In Frauenknecht’s meta-analysis including 23 randomized studies and 1300 patients, with a high level of evidence, OFA decreased the rate of postoperative nausea and vomiting, but had no effect on postoperative pain. The study did not evaluate the rate of respiratory complications [9]. In the recent meta-analysis by Salomé et al., conducted on 2209 patients in 33 randomized controlled trials, the OFA technique showed a reduction in PONV and pain in PACU, but had no effect on postoperative pain or opioid consumption at 48 h. This study did not find more hemodynamic complications in the OFA group [10]; finally, in another recent meta-analysis, Olausson et al. found that OFA significantly reduced adverse postoperative events in many common interventions, such as gynecological, upper gastrointestinal, and breast surgeries [8].

The comparison between studies is complex because each trial has its own OFA protocol for the reduction or even suppression of intraoperative opioids, and the judgment criteria are not standardized.

To our knowledge, this study is the first to focus on ORA in major cervicofacial oncologic surgery, where patients are intrinsically at high risk of pain due to the tumor localization, but also because of multi-site nature of the surgery. This study specifically informs us concerning an understudied, yet morbid, population regarding anesthesia.

This study has some limitations. The first is its retrospective nature and its limited sample size, which results in a lack of ability to detect differences between the two groups. The use of remifentanil, even in low target concentrations, might be questionable; however, anesthesia providers at the time of the study were not familiar with opioid reduced anesthesia and preferred to have a back up “ready to use” opioid in case of severe sympathetic response to nociceptive stimulation. Additionally, the inclusion of patients with preoperative opioid consumption could also be questioned; as this was not a randomized study, but a retrospective case-controlled investigation, we preferred to check the effect of this protocol on these patients as well. In a previous study, we described an increase of 40% regarding opioid requirements in these patients undergoing major cancer surgery, and we hypothesized that any opioid-saving effect would be beneficial to these patients [24]. Finally, major cervicofacial surgery includes multiple types of surgery, and the most complex types are those with free flap reconstruction. Usually, free flaps are harvested from a distant site, such as the fibula, quadriceps, or scapula, and intense postoperative pain can emanate from the harvest site. In addition to those at the cervical site, we believe that regional blocks performed by anesthesiologists are truly beneficial in this category of patients (in contrast to catheters placed by surgeons) [18]; therefore, since this was a retrospective study, it was not permitted to exclude patients who had peripheral regional blocks. However, there was no cervical site block in any case (since these blocks are not performed in our institution); therefore, the pain emanating from the cervicofacial site is constant and significant. The groups were not totally comparable, as there were patients in the control group who were more critical and who underwent more complex procedures, which limited the interpretation of the results. This initial difference between the 2 groups can be explained by an important selection bias, since, despite the broad inclusion criteria, patients in the ORA group were pre-selected based on the absence of comorbidities as part of the introduction of a new protocol. Nevertheless, we tried to adjust the two groups for the main endpoints by using additional linear regression and multivariate analysis, and we did not find differences in comparison to our initial results. Intraoperative nociception was not monitored, with opioid administration left to the discretion of the clinician based on the hemodynamic response to nociceptive stimuli that was potentially minimized by using α-2 agonists. The monitoring of nociception by a pupilometer or a nociception monitor was not possible due to the sympatholytic mechanism of action of dexmedetomidine and the surgery site. Unfortunately, we did not have other means of measuring nociception, which was not routinely monitored at the time of the study.

To date, there are few randomized controlled studies using large numbers to validate opioid-reduced anesthesia. The comparison of existing studies is complex because different endpoints are studied—postoperative pain, hypoxemia, respiratory complications, PONV—and each OFA or ORA protocol is specific to the anesthesia team that implements it. The use of dexmedetomidine in these protocols is not without risk, and the POFA study [22] suggests more caution regarding its use in the face of severe bradycardia, which led to its premature withdrawal.

The opioid health crisis alone does not justify denigrating one of the historical pillars of anesthesia and postoperative analgesia. It is important to contextualize the use of opioids, which is necessary in 3% of postoperative patients after 3 months [2], while chronic postoperative pain persists in 12% of patients [5].

To date, postoperative analgesia, although a key issue in perioperative management, has not been optimized. Multimodal analgesia, although its effectiveness has been demonstrated in the literature, is far from being ubiquitous. In Ladha’s study, only 56% of the patients received non-opioid multimodal analgesia postoperatively [25]. Similarly, it is estimated that only 3% of patients benefit from regional analgesia compared to the 25% who are eligible [26]. The reduction in opioids would be the logical consequence of a better postoperative analgesic management. In the absence of a “one size fits all” policy, it would be judicious to adapt anesthesia and postoperative analgesia to each patient according to their risk factors and the surgery that awaits them by favoring multimodal anesthesia and analgesia, which would allow for opioid sparing.

Cervicofacial oncologic surgery is an excellent example of the complexity of perioperative analgesic management, and it could be the target of a multimodal anesthesia that could include α-2 agonists as adjuvants, but not replacements, for opioids.

## 5. Conclusions

Except for in the first postoperative hours, this retrospective study did not find a significant improvement in the management of post-operative analgesia after the implementation of an opioid-reduced anesthesia protocol in major cervicofacial oncologic surgery. Prospective studies are necessary regarding this type of complex surgery to better manage postoperative pain in these patients.

## Figures and Tables

**Figure 1 jcm-12-00904-f001:**
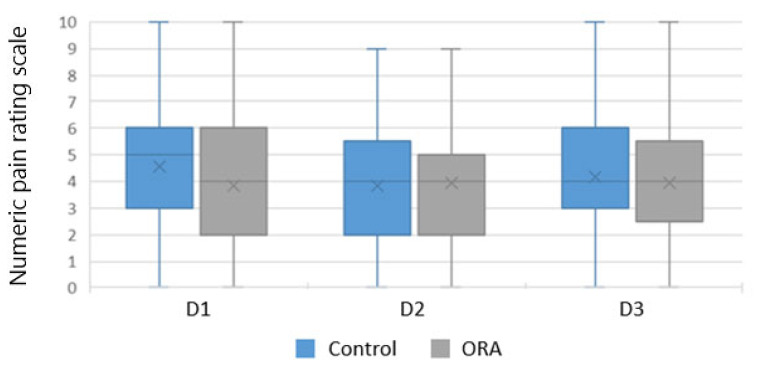
Pain assessment over the first 72 h. ORA: opioid reduced anesthesia; D: postoperative day.

**Figure 2 jcm-12-00904-f002:**
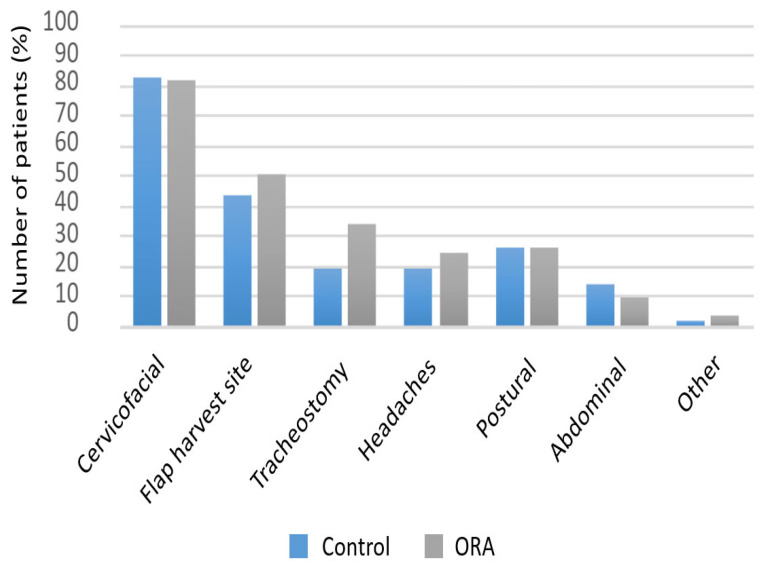
Pain location; ORA: opioid reduced anesthesia.

**Table 1 jcm-12-00904-t001:** Baseline characteristics of patients.

Characteristics	Total	Control *n* = 86	ORA *n* = 86	*p* Value
Female sex, *n* (%)	68 (39.6)	33 (38.4)	35 (40.7)	0.87
Age (years), mean (min-max)	57.5 (19–91)	59.2 (19–91)	55.8 (20–77)	0.08
ASA score I and II, *n* (%)	131 (76.2)	58 (67.4)	73 (84.9)	0.01
Comorbidities				
High blood pressure, *n* (%)	51 (29.7)	31 (36)	20 (23.3)	0.09
Diabetes, *n* (%)	16 (9.3)	12 (14)	4 (4.7)	0.06
Cardiovascular disease *, *n* (%)	9 (5.2)	8 (9.3)	1 (1.2)	0.04
COPD, *n* (%)	76 (44.2)	45 (52.3)	31 (36)	0.04
Sleep apnea syndrome, *n* (%)	8 (4.7)	2 (2.3)	6 (7)	0.27
Obesity, *n* (%)	18 (10.5)	9 (10.5)	9 (10.5)	0.99
Malnutrition, *n* (%)	25 (14.5)	18 (20.9)	7 (8.1)	0.029
Chronic pain treated by opioids, *n* (%)	33 (19.2)	18 (20.9)	15 (17.4)	0.71
Previous cancer, *n* (%)	52 (30.2)	36 (41.9)	16 (18.6)	0.002
Surgery				
Median duration, hours [min-max]	10 [4–16]	10 [4–14]	10 [5–16]	0.58
Free flap reconstruction, *n* (%)	158 (91.9)	84 (97.7)	74 (86)	0.01
Tracheostomy, *n* (%)	153 (89)	82 (95.3)	71 (82.5)	0.4
Mandibulectomy, *n* (%)	62 (36)	35 (40.7)	27 (31.4)	0.26
Glossectomy, *n* (%)	35 (20.3)	15 (17.4)	20 (23.3)	0.44
Maxillectomy, *n* (%)	31 (18)	13 (15.1)	18 (20.9)	0.67
Pharyngectomy, *n* (%)	41 (23.8)	26 (30.2	15 (17.4)	0.07
Laryngectomy, *n* (%)	18 (10.4)	6 (7)	12 (14)	0.21
Other **, *n* (%)	27 (15.7)	11 (12.7)	16 (18.6)	0.81

* Cardiovascular disease, including coronary artery disease, peripheral arterial disease, and carotid atherosclerosis. ** Other surgery, including parotidectomy, ventriculoplasty, and ethmoidectomy crico-hyoido-ethmoido-pexy. ASA: American Society of Anesthesiology; COPD: chronic obstructive pulmonary disease; ORA: opioid reduced anesthesia.

**Table 2 jcm-12-00904-t002:** Intraoperative events.

	Total *n* = 172	Control *n* = 86	ORA 86	*p* Value
Propofol (mg/kg), median [IQ 25; 75]	2.46 [2.03; 2.86]	2.42 [2.01; 2.85]	2.5 [2.06; 2.86]	0.263
Remifentanil for intubation, *n* (%)	135 (78.5)	85 (98.8)	50 (58.1)	<0.001
Dexmedetomidine bolus (μg/kg), median [IQ 25; 75]	-	-	0.4 [0.3; 0.5]	-
Maintenance				
Sevoflurane, *n* (%)	162 (94.2)	80 (93)	82 (95.4)	0.746
Desflurane, *n* (%)	8 (4.7)	5 (5.8)	3 (3.5)	0.720
TIVA Propofol, *n* (%)	2 (1.2)	1 (1.2)	1 (1.2)	0.999
Minimal infusion rate of dexmedetomidine (μg/kg/h), median [IQ 25; 75]	-	-	0.2 [0.2; 0.4]	-
Maximal infusion rate of dexmedetomidine (μg/kg/h), median [IQ 25; 75]	-	-	0.8 [0.7; 1]	-
Regional anesthesia, *n* (%)	111 (64.5)	48 (55.8)	63 (73.3)	0.025
Perineural catheterization, *n* (%)	93 (54.1)	56 (65.1)	37 (43)	0.006
Fluid infusion (mL/kg/h), median [IQ 25;75]	10 [8; 12]	10 [8;12]	10 [8;12]	0.987
Intraoperative transfusion, *n* (%)	88 (51.2)	58 (67.4)	30 (34.9)	<0.001
Urine output (mL/kg/h), median [IQ 25; 75]	1.6 [0.8;3]	1.3 [0.8; 3]	2.1 [1.2; 3.3]	0.042

ORA: opioid reduced anesthesia; IQ: interquartile; D: postoperative day.

**Table 3 jcm-12-00904-t003:** Secondary endpoints: opioids side effects and intraoperative hemodynamic tolerance.

Secondary Endpoints	Total *n* = 172	Control *n* = 86	ORA *n* = 86	*p* Value
Postoperative opioid side effects				
Hypoxemia, *n* (%)	103 (59.9)	60 (69.8)	43 (50.0)	0.013
Additional PAV or HFO, *n* (%)	21 (12.2)	16 (18.6)	5 (5.8)	0.018
PONV, *n* (%)	33 (19.2)	16 (18.6)	17 (19.8)	1.000
Ileus, *n* (%)	43 (25)	20 (23.3)	23 (26.7)	0.723
Acute urinary retention, *n* (%)	20 (11.6)	9 (10.5)	11 (12.8)	0.813
Delirium, *n* (%)	11 (6.5)	6 (7.1)	5 (5.8)	0.764
Intraoperative hemodynamic tolerance				
Bradycardia, *n* (%)	6 (3.5)	2 (2.3)	4 (4.7)	0.682
Norepinephrine infusion, mean ± SD (mg/h)	0.2 ±0.2	0.1 ±0.2	0.2 ±0.2	0.044

HFO: high-flow oxygen; PAV = pressure-adjusted ventilation; ORA: opioid reduced anesthesia; SD: standard deviation. PONV= postoperative nausea and vomiting.

## Data Availability

Data are available on demand (L.B.).

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
