# Peer review of "Opioid Reduced Anesthesia in Major Oncologic Cervicofacial Surgery: A Retrospective Study"

_jcm, 2023, doi:10.3390/jcm12030904_

Round 1

Reviewer 1 Report

The authors compared an opioid sparing technique to a non-opioid sparing technique.  The authors should further highlight the difference in groups and its confounding.  The introduction is too long.  The immunomodulation may be deleted since this aspect was not studied.  The inclusion of remifentanil in the non-opioid sparing technique should be further clarified especially given the research on the hyperalgesic effect.  The inclusion of regional analgesia for certain patients is confusing and should be further clarified.  The difference in respiratory events must be further clarified.  What were the criteria for the institution of non-invasive ventilation and high flow oxygen?  What is an isolated episode of hypoxemia?  What was the influence of bias given the introduction of opioid reduced anesthesia?

Author Response

The authors compared an opioid sparing technique to a non-opioid sparing technique.  The authors should further highlight the difference in groups and its confounding. 

 Thank you for your suggestion , we  addressed this issue  by performing linear regression which was performed by Arnaud Pagès  from our dpartement of biostatistics  who has a PhD in biostatistics , and added results accordingly. No difference in comparison to the primary result was obtained  after adjustements 

The total report of statistics is available on demand (35 pages)

The introduction is too long.  The immunomodulation may be deleted since this aspect was not studied. 

Thank you for your suggestion we shortened the introduction and removed unnecessary sentences for immunomodulation  as suggested .

The inclusion of remifentanil in the non-opioid sparing technique should be further clarified especially given the research on the hyperalgesic effect. 

Thank you for your comment , this protocol was initiated almost three years ago, anesthesia providers were not familiar with OFA at the time of the study   and preferred to have  a back up ready to use opioid  just in case .

we added a miniparagraph about this issue in the discussion section 

Hyperlagesia  is largely underdebate , in our group of patients we mostly encounter it in chronic pain patients In our practice we have found that chronic pain patients increase their demand of remifentanil by 40% : 

  • DOI: 10.5055/jom.2017.0390

The inclusion of regional analgesia for certain patients is confusing and should be further clarified. 

Major cervicofacial surgery include different types of surgery the most complex ones are those with free flap reconstruction .  usually free flaps are harvested on a distant site like fibula or quadriceps , or scapula ., intense postoperative pain can emanate from the harvest site in addition to cervical site , indeed we published recently about this subject (DOI: 10.3390/jcm11216384) ,  we believe regional blocks performed by anesthesiologist  are truely beneficial   in this category of patients  (in contrast to catheter placed by surgeons), therefore since this was a retropective study , it was not permitted  to exclude patients  who had  peripheral regional block , however    there was no cervical site block in any case (since these blocks are not performed in our institution) , the pain emanating from the cervicofacial site is constant and important. , we added a sentence to explain this issue.

This was a retrospective study was not designed to see a specific effect in a prospective methodology , in fact the objective was to assess  the use and opioid sparing effect  DXM in all type of major cerviocofacial surgery , we added this paragraph in the discussion section 

The difference in respiratory events must be further clarified.

We detailed as requested  the respiratory event,  i

What were the criteria for the institution of non-invasive ventilation and high flow oxygen?

Patients with hypoxemia or desaturation less than 95% necessitating more than 3L of oxygen or other type of mild respiratory complications such as atelectasis could benefit from high flow nasal oxygen therapy or intermittent pressure assisted ventilation support  if there was not  a nead for continuous  invasive ventilation. The intensivist in charge however could decide at any time to transfer the patient to medical intensive care unit for more respiratory support. 

  What is an isolated episode of hypoxemia?  What was the influence of bias given the introduction of opioid reduced anesthesia?

Thank you for your suggestion , we deleted isolated in the text at it was inappropiate 

As you suggested we adjusted the two groups  for comparison , but we do not believe these episodes of hypoxemia are directly related to the ORA protocol , as no signicant difference in  morphine requirement was observed after day 1, and the other confounding factor is the preemptive pressure adjusted ventilation which was executed similarily in both groups

Reviewer 2 Report

1) In the middle of 4th paragraph, please note the definition of hyperalgesia seems not accurate, hyperalgesia and allodynia are different.  

2)  Why didn't exclude those patients regularly use opioids againts chroinc pain before the surgery. Although authors made a comparison of the opioids use in Table-1

3) The preoperative recongnition deficits undoubtfully have impacts on POD. How authors exclude this? 

Author Response

1) In the middle of 4th paragraph, please note the definition of hyperalgesia seems not accurate, hyperalgesia and allodynia are different.  

Thank you for this correction , we apologize and  made the change.

Hyperalgesia refers to an increase sensitivity to feeling pain from a stimulus that usually provokes pain  

2)  Why didn't exclude those patients regularly use opioids againts chroinc pain before the surgery. Although authors made a comparison of the opioids use in Table-1

Thank you for this comment , this study was designed to assess if DXM could help reducing opioid use  in major cervicofacial surgery , we believe chronic patients with opoid use should also be part of the targeted population and may benefit from this protocol we added a mini paragraph in the discussion section 

,  as this was not a randomized study but a retrospective one we preferred to check  the effect of this protocol also on these patients., In  previous study  we described an increase in 40% of opioid requirement in these patients having major cervicofacial surgery and any opioid and we thought any opioid saving effect would be beneficial to these patients 

https://www.ncbi.nlm.nih.gov/pubmed/28953314

3) The preoperative recongnition deficits undoubtfully have impacts on POD. How authors exclude this? 

You are right , in general when a patient have preoperative  cognitive disorder , they are not sceduled for major surgery unless the cognitive dysfunction is resolved  or in case of an emergency situations,  in these later  scenario it is mainly airway obstruction and there is a shift to palliative care not a  cured treatement option., we added a miniparagraph in the method 

The indication of these major surgeries   was decided in a multidisciplinary committee; patients with cardiovascular, respiratory instability and cognitive disability and vital emergency situations were excluded until improvement and stabilization.